# Grain Legumes and Fear of Salt Stress: Focus on Mechanisms and Management Strategies

**DOI:** 10.3390/ijms20040799

**Published:** 2019-02-13

**Authors:** Muhammad Nadeem, Jiajia Li, Muhammad Yahya, Minghua Wang, Asif Ali, Andong Cheng, Xiaobo Wang, Chuanxi Ma

**Affiliations:** 1School of Agronomy, Anhui Agricultural University, Hefei 230036, China; rananadeem.aaur@yahoo.com (M.N.); lijia6862@ahau.edu.cn (J.L.); minghuawang.ahua@gmail.com (M.W.); genetics08248@gmail.com (A.C.); machuanxi@ahau.edu.cn (C.M.); 2School of Life Sciences, Anhui Agricultural University, Hefei 230036, China; yahyapbg@gmail.com (M.Y.); aligenetics08@gmail.com (A.A.)

**Keywords:** legumes, salt stress, tolerance, genomics, speed breeding, CRISPR-cas9

## Abstract

Salinity is an ever-present major constraint and a major threat to legume crops, particularly in areas with irrigated agriculture. Legumes demonstrate high sensitivity, especially during vegetative and reproductive phases. This review gives an overview of legumes sensitivity to salt stress (SS) and mechanisms to cope with salinity stress under unfavorable conditions. It also focuses on the promising management approaches, i.e., agronomic practices, breeding approaches, and genome editing techniques to improve performance of legumes under SS. Now, the onus is on researchers to comprehend the plants physiological and molecular mechanisms, in addition to various responses as part of their stress tolerance strategy. Due to their ability to fix biological nitrogen, high protein contents, dietary fiber, and essential mineral contents, legumes have become a fascinating group of plants. There is an immense need to develop SS tolerant legume varieties to meet growing demand of protein worldwide. This review covering crucial areas ranging from effects, mechanisms, and management strategies, may elucidate further the ways to develop SS-tolerant varieties and to produce legume crops in unfavorable environments.

## 1. Introduction

To meet the challenge of feeding a continuous growing population and there will be an urgent need to enhance productivity around 87% or even more of what we are producing at present especially cereals and legumes by 2050 [1]. However, major abiotic constraints, which includes drought, heat, cold, and salinity critically threatens productivity and causes major yield loss in large areas [2,3,4]. Among these, salt stress (SS) is one of the major constraint to profitable crop production [5] and is likely to increase due to environmental changes and as a result of various irrigation malpractices. SS negatively influences 60 million hectares, or round about 20% of the total irrigated land area in the world [6]. Salinity can be termed as abiotic stress and comprises all the problems due to salts primarily by an abundance of sodium chloride (NaCl) from irrigation or natural accumulation [7].

Legumes belong to family Fabaceae and are a nourishing and low-cost food. Legumes are an excellent source of proteins and edible oil and can play a significant role in meeting food and oil needs in the present situation of global climate change. Legume crops also are a major source of carbohydrates, isoflavones vitamins, fiber, and minerals for human and animals. Crop rotation that comprises legumes is an effective approach to replenish fertility of the soil and enhances yields due to the nitrogen fixation [8]. Worldwide grain legumes occupy 12–15% of arable land to provide 33% of dietary protein and produce 27% of major crop production [9].

In legumes, salinity effects overall plant growth and development [10,11], by disturbing the nutritional imbalances, the complex interaction of hormones, osmotic effects and specific ion toxicity [12,13]. SS also disturbs composition of grains and grain yield [14,15]. For instance, increased necrosis and chlorosis of leaves observed under salinity, which leads to leaf senescence and decrease in photosynthesis in grain legumes [16,17]. The influence of salinity stress on several other crops has been previously reviewed [18,19], but no comprehensive study is available on the sensitivity, mechanisms, and approaches of salinity stress tolerance in legume crops. Our review about effects, mechanisms, and management strategies may lead to the development of salt-tolerant cultivars under SS.

## 2. Legumes Sensitivity to Salt Stress (SS)

Salinity has devastating impacts on plants by affecting germination and growth, reproductive stage, and the ability to biological nitrogen fixation in grain legumes [20]. Salinity affects vital physiological functions, nutritional imbalances, and hormonal regulation [21,22], reduced carbon fixation [10], flower abortion, and reduced numbers of flowers and pod set, and eventually impedes crop production [17]. Salinity is becoming one of the main threat to crop productivity and has an inimical impact on plants (Figure 1).

Under SS, plant growth is affected at two phases; early phase (osmotic phase) and late phase (ionic phase) [23]. The early response of plants under saline conditions is due to the salt outside the root zone whereas SS at the late phase is the consequence of the toxic effect of salt accumulation inside the plant [24]. Seed germination is a critical stage during the plant life-cycle. Salinity affects seed germination by the toxic effect of ions on the seed embryo and inhibiting the water uptake [25]. Shu et al. [26] reported that SS inhibits soybean seed germination by decreasing the gibberellin (GA)/abscisic acid (ABA) ratio. Kumar et al. [27] reported that salinity reduces germination and seedling growth of soybean. Ma et al. [28] observed that NaCl negatively affects seed germination in alfalfa. Haileselasie and Teferii, [29] found that SS severely affects germination and growth of chickpea. Similarly, the 50% reduction in seed germination observed due to the increasing level of salinity from 0–180 mM NaCl in *Phaseolus vulgaris* L. [30]. 

In legumes, higher sensitivity to SS is observed during seedling and developmental stages than the germination stage [31]. Salinity limits the growth of several legumes, including mungbean [32], soybean [33], lentil [34], and faba bean [35]. These growth limitations are often attributed to the decline in water potential of tissue, indicating less availability of water to cells [36], which results in closure of stomata, declined photosynthesis, and inhibited growth [37]. The chlorophyll content, as an indicator of photosynthetic activity, normally reduced under salinity in soybean plant [38]. Ning et al. [39] reported decline in photosynthesis and growth of soybean under salinity. A salinity-induced decline in photosynthesis is not only as a result of stomatal limitations, but also due to non-stomatal factors including reduced photosynthetic pigments, chlorophyll fluorescence, and ultrastructural damage [40,41]. The reduced plant growth under salinity is often associated with reduced photosynthetic activity [42]. Zawude and Shanko, [43] found that chickpea is sensitive to SS, particularly during the period of early stages of growth and development. Demir and Arif, [44] drew attention to the fact that salinity adversely affected root length compared to shoot length.

Salinity affects the competitive nutrient uptake, accumulation, and transport in plants. Under salinity, the abundance of Na^+^ and Cl ^−^ ions concentrations in the rhizosphere will cause nutritional imbalance in plants as these ions interfere with other nutrients, including nitrogen (N), phosphorus (P), potassium (K), boron (B), calcium (Ca), zinc (Zn), copper (Cu), magnesium (Mg), and iron (Fe) [21,22,45]. Previous investigations have confirmed that the harmful effects of SS on the plant may occur through an ionic imbalance, particularly of K^+^ and Ca^2+^. According to Greenway and Munns, [46], plants must maintain relatively higher concentrations of K^+^ and Ca^2+^ if they are to survive successfully under SS environment. Higher salt contents decrease the concentrations of Ca^2+^, K^+^, Mg^2+^ and other cations, which play a vital role in the photosynthetic activity of plant [47]. For instance, a significant reduction of Na^+^/K^+^ ratio observed in mungbean [48], and chickpea [36], due to competitive uptake of Na^+^ and K^+^ ion flux, resulting in deficiency of K^+^ and significant yield losses [49]. In soybean, salinity stress reduced the accumulation of Ca^2+^, K^+^, and Mg^2+^ in the leaves [50]. In addition, SS has an adverse impact on nodulation and final yield. Early studies uncovered that SS interferes with N uptake and biological N fixation, which limits N supply in legumes [51]. Nodules are responsible for N fixation in legumes; however, the mechanism of nodulation is extremely sensitive to SS. For instance, previous studies revealed that SS substantially affect the activity and density of nodules in pigeon pea [15] and faba bean [51,52] and leads to premature nodules senescence [53]. Legumes are highly sensitive to SS which leads to sever yield losses (Table 1). Khan et al. [54] reported a significant reduction in yield and yield-related traits (number of pods and seed weight) under SS in soybean. Similarly, Ghassemi-Golezani et al. [55] noted that all yield-related traits were equally responsible for final yield reduction under SS in soybean. In the case of chickpea, SS-induced a decrease in final yield due to shriveled seeds and a reduction in grain weight by 20% [56,57]. Ahmed, [58] observed fewer grains per pod and lower grain weight under salinity stress in mungbean. Recently, Farooq and coworkers review the diverse effects of SS on legume yield. They reviewed that SS leads to yield losses (12–100%) in various legumes [59]. More importantly, a growing number of reports showed that SS adversely affects the composition and quality of grain legumes [15]. For instance, SS significantly reduced protein contents in mungbean, chickpea, and faba bean due to a decline in NO_3_ supply from the soil [52,55,60]. In soybean, SS affects grain oil and protein contents; the protein and oil contents decreased with increasing salt concentrations compared with control [55]. It has been reported that higher concentrations of salt decreased the amino acid, carbohydrates, polysaccharides, and protein contents in grains of mungbean [61]. In conclusion, salinity disturbs overall plant growth in legumes by influencing seed germination, photosynthesis, nutrient uptake and nutritional imbalance, and final yield.

## 3. Tolerance Mechanisms

To increase the yield of legumes under SS, it is imperative to first understand tolerance mechanisms. Plants are evolved with different adaptations including ion homeostasis, compatible solute accumulation and osmotic protection, antioxidant regulation, and hormonal regulation. Research advances elucidating these mechanisms are discussed below.

### 3.1. Ion Homeostasis and Salt Tolerance

In plants, ion homeostasis is an imperative mechanism of cells under SS. It helps to maintain lower concentrations of Na^+^ and higher concentrations of important ion K^+^ [25,63]. Ion flux regulation of intracellular Na^+^ and K^+^ is fundamental for the activities of different enzymes in the cytosol as well as for the regulation of cell volume and maintenance of membrane potential [63]. Under SS, plants eliminate unnecessary salts from the cytosol by primary active transport along with secondary transport to maintain the cytosolic levels of Na^+^ and K^+^ [63,64]. Under salinity, to sustain ion homeostasis plant cell regulate these cation transporters in the tonoplast and plasma membranes [64]. 

The Na^+^ compartmentalization or exclusion from vacuole, providing a fundamental feature to avoiding the lethal effects of Na^+^ in the cytosol of the cell [64]. Variation in ions distribution exists between species and cultivars of legumes, particularly the ratio of K^+^/Na^+^ in the cytosol [52]. For instance, Mishra et al. [9] reported that the mash bean cultivar (T-44) has unique K^+^/Na^+^ transporters which assist in maintaining lower level of intracellular Na^+^. While in pigeon pea, Subbarao et al. [65] reported a decreased Na^+^/Ca^2+^ ratio under salinity regulates the uptake of K^+^ which leads to improve Na^+^/K^+^ ratio. It has been observed that a mechanism for salinity tolerance comprises exclusion of Na^+^ and Cl^−^ from shoots, maintenance of higher Na^+^/K^+^ ratio in the shoots, and increased uptake of K^+^ in pigeon pea [66]. The ion sequestration in older tissues and the elimination of Na^+^ and Cl^−^ through roots may help to protect younger leaves and reproductive organs [23]. Turner et al. [67] observed that SS in chickpea is associated with high concentrations of Na^+^ in seeds and young leaves, but in older tissues, no association was found. Under salinity, exclusion of Na^+^ from the transpiration stream into the xylem by exchange with K^+^ at the xylem/symplast boundary of the roots [68]. The exchange is assisted by antiporters (K^+^/H^+^ and Na^+^/H^+^), boosted by H^+^-ATPase, dependent on anion permeability, and further boosted by higher concentrations of apoplastic K^+^ [69]. In soybean, the activity of H^+^-PPase and H^+^-ATPase in the tonoplast of salt-tolerant cultivar improved under salinity, relative to a sensitive cultivar [70].

Increasing evidence revealed the role of a salt overly sensitive (SOS) pathway in ion homeostasis and salt tolerance [63,71]. The SOS pathway comprises of three noteworthy proteins, SOS1, SOS2, and SOS3 (Figure 2). The *SOS1* encodes Na^+^/H^+^ antiporter and is essential in regulating Na^+^ efflux at the cellular level. It also assists in the movement of Na^+^ from root to shoot in the plant. Overexpression of *SOS1* improves salt tolerance in plants [72]. The antiporter SOS1 assists in the removal of excessive Na^+^ from roots and involves long-distance Na^+^ transport in xylem. The SOS1 homolog has been observed in all crop plants and possibly plays a fundamental role in legumes under salinity [9]. The *SOS2* encodes a serine/threonine-specific protein kinase, is stimulated by salinity stress elicited Ca^2+^ signals. The *SOS2* comprises of a C-terminal regulatory domain and an N-terminal catalytic domain [73]. The *SOS3* protein is a myristoylated Ca^+^ binding protein and has a myristoylation site at N-terminus. This myristoylation site plays a fundamental role in regulating salinity tolerance [74]. In conclusion, the removal of excess Na^+^ and Cl^−^ ions and their compartmentalization into old tissues or vacuoles, is an important feature for salinity stress tolerance.

### 3.2. Compatible Solute Accumulation and Osmotic Protection

To cope harmful osmotic effects during phase I of SS, crop plants adopt a fundamental strategy termed as osmoregulation [25]. In the process of osmoregulation, the cell water potential declines without any decrease in turgor. Osmotic stress adjustment occurs through uptake of ions and synthesis of solutes like polyols, sugars, amides, amino acids, quaternary ammonium compounds and proteins [75]. Organic solutes and inorganic ions are involved in osmotic stress adjustment in the plant under SS, but their comparative functions are different between genotypes and species. The solutes which are not toxic even at higher concentration and involved in osmotic adjustment are called compatible solutes. These compatible solutes are hydrophilic compounds with low molecular weight and no net charge at physiological pH and also termed as osmoprotectants [68]. El Sayed [22] observed that salt-tolerant bean plants had less protein and high proline and amino acids contents than a salt-sensitive broad bean. Similarly, salt-tolerant mash bean and faba bean had higher concentrations of leaf proline under salt stress than the control [52,76]. Additionally, pigeon pea and mungbean accumulated more glycine betaine (GB) under salinity [75,77]. In chickpea, proline, soluble sugars, choline, and GB accumulation helped to maintain photosynthetic pigments and improved plant growth and development under salinity conditions [78]. In soybean, the regulation of trigonelline, proline, and potassium concentrations improve osmotic adjustment in plants [79]. Similarly, the accumulation of amino acids, reducing sugars, and ascorbic acid involved in the osmotic adjustment in pea plants under salt stress [80]. In conclusion, proline and GB are the key osmolytes involved in osmoregulation and diminish the effects of osmotic stress in grain legumes.

### 3.3. Antioxidant Regulation of Salinity Tolerance

Like other stresses, salinity stress might uncouple various metabolic pathways and several enzymes, which results in the accumulation of lethal and undesirable reactive oxygen species (ROS), such as hydrogen-peroxide (H_2_O_2_), hydroxylradical (^•^OH), superoxide-radical (O_2_^•−^), and singlet-oxygen (O_2_). These ROS damage cells, along with different proteins, nucleic acids, and membrane lipids, which lead to oxidative stress. Salt-tolerant legumes have an antioxidant defense system (Figure 3), to deal ROS by antioxidants activities [25]. Plants have a diverse array of non-enzymatic antioxidants, such as glutathione, carotenoids, tocopherols, ascorbic acid, flavonoids and flavones, and certain antioxidant enzymes, such as glutathione reductase (GR), glutathione peroxidases (GPX), ascorbate peroxidase (APX), superoxide dismutase (SOD), glutathione S-transferases (GST), monodehydroascorbate reductase (MDHAR), and catalase (CAT) to remove ROS [81,82]. Under salinity stress, it has been observed that the SOD, MDHAR, DHAR, GR, and APX activities increased in a salt-tolerant cultivar of pea [83]. Similarly, Yasara et al. [84] reported that SS enhanced APX and CAT activities in a salt-tolerant common bean cultivar (Gevas sirsk 57). In conclusion, increased antioxidant activities in legumes help to improve salinity tolerance by protecting from oxidative stress. 

### 3.4. Hormone Regulation of Salinity Tolerance

Plant hormones (phytohormones) are chemicals that control and regulate all aspects of growth and development of plants. Major plant hormones are categorized into two categories growth promoters (gibberellins, cytokinins, and auxins) and growth retardants (ABA and ethylene). Under saline conditions, deviations in ABA and ethylene concentration are often observed. SS leads to the ABA up-regulation, which in turn elicits diverse adaptive responses in plants [85]. ABA pathway comprises the PYR, PYL, RCAR proteins complex, protein phosphatases 2C (PP2C), ABF transcription factors and SnRK2 protein kinases (Figure 3) [86]. In legumes, ABA is important for various stress responses, including metabolic changes, stomatal closure, and regulation of stress responsive genes. Enhanced concentration of ABA in other tissues and leaves not only disturbs stomatal activity but improves the production of stress proteins and salt adaptation by osmotic adjustment [87]. For instance, *Lupinus albus* L. closed stomata in response to ABA produced locally then to ABA synthesized by the roots and transported to leaves [88]. Under SS, the synthesis of ethylene and 1-aminocycloprane-1-carboxylic acid (ACC) precursor in nodules and roots activated senescence of leaves but was not associated with relatively better growth in faba bean and chickpea [82]. Thus, salinity changes the concentrations of hormones which induce alterations in osmotic adjustment, photosynthesis, and plant growth.

## 4. Management Strategies to Improve Salt Tolerance in Legumes

Salt-tolerant plants grow and complete their life span on a substrate that comprises higher concentrations of soluble salt. Plants that can perform well on higher levels of salt in the rhizosphere and grow well are termed as halophytes. Salt tolerance is a complex trait comprising several interacting properties. To meet the challenge of feeding a continuous growing population, scientists and breeders are continuously looking for strategies to enhance crop productivity. The following sections will briefly review the strategies for improving SS tolerance of legumes under saline conditions. 

### 4.1. Agronomic Strategies to Reduce SS

Salinity stress develops due to salt accumulation through soil chemical properties and irrigated water. Under saline conditions due to Na^+^ and Cl^−^ toxicity, the ratios of Na^+^/Ca^2+^, Na^+^/K^+^, Cl^−^/NO^3−^, and Ca^2+^/Mg^2+^ increased in the soil. The imbalance in ion concentration effects plant growth, yield, and metabolic and physiological components [89,90]. Legume growth can be managed and improved by adopting certain agronomic strategies (nutrient management and water management) and, thus, will improve plant growth, soil health, and input use efficiency under saline conditions.

#### 4.1.1. Reclamation of Salt-Affected Soils by Nutrients Management

Under saline conditions, nutrients imbalance was observe due to Na^+^ and Cl^−^ toxicity and deficiency of major (N, P, K^+^) and minor (Ca^2+^, S, Mn, Zn^2+^) nutrients in several crops [91,92]. Suitable fertilizer applications or nutrient management on crop plants is a very practical way for alleviating salt injury [93,94]. In order to manage nutrients in salt-affected soils, the main approach is the application of gypsum (CaSO_4_·2H_2_O), which is a major source of Ca^2+^, to improve soil water infiltration and for reclamation of Na^+^ toxic soil. Under saline conditions, application of 100% CaSO_4_·2H_2_O, the combination of H_2_SO_4_+farm yard manure+CaSO_4_·2H_2_O, CaSO_4_·2H_2_O+chiseling+farm yard manure, humic acid (HA), and pyrite, have improved plant growth, soil properties, and yield in several crops [95,96]. Matuszak-Slamani et al. [97] reported that application of HA improves growth and development of soybean under saline soil environment. Lawson et al. [98] studied the effects of compost on growth and nodulation of the kidney bean, soybean, and alfalfa. They observed that growth and nodulation were improved by compost under saline conditions. Sun et al. [99] observed that biochar application to the saline soils can reduce NH_3_ volatilization, keep N retention, and decrease N leaching, which is beneficial for sustainable use of salt-affected soils. Additionally, exogenous osmoprotectants combined with compost will effectively solve salinity problem and are a good strategy to increase salinity resistance of soybean in the drylands [100]. In faba bean, soil organic matter improves plant growth due to the gradual release of certain nutrients in salt-affected soils [101]. Under low SS, the application of urea to N-deficient soil improved growth and productivity in chickpea [102]. Under mild saline conditions, enhancing K^+^ concentrations improved growth, water relations, and productivity of mungbean [103]. Guo et al. [104] reported that increasing NO^3−^ supply for crop plants under SS can significantly reduce Cl^−^ content and alleviate crop salt injury or enhance salt tolerance in soybean. It has been observed that the application of micronutrients improves growth and nutrient uptake either before or after the salinization treatment. The foliar application of micronutrients could induce an increase in SS tolerance in faba bean [105]. Similarly, it has been reported that foliar spraying with Zn or Fe chelate improves kidney bean SS tolerance, during early growth stages [106]. Exogenous application of Si mitigated the harmful effects of Na^+^ and Cl^−^ in common bean, cowpea, faba bean under saline conditions [11,107,108]. To mitigate the adverse effects of SS on the plant, we can also adopt some other agronomic approaches, such as subsoiling (20–150 cm apart and 50 + 5 cm crosswise, furrows), sanding, use of fresh water, application of inorganic and organic fertilizers, and deep tillage [92,96]. In conclusion, the application of potash and nitrogenous fertilizers, and micronutrients foliar application may help to enhance growth and productivity under saline conditions.

#### 4.1.2. Reclamation of Salt-Affected Soils by Water Management

Decline in yield depends on crop growth, concentrations of salts in irrigated water and soil, and climatic conditions. The approach used to eliminate soluble salts from rhizosphere is termed as the reclamation of saline soils. Reclamation of salt-affected soils for sustainable usage is a matter of great concern [99]. Irrigation water with higher pH, residual sodium carbonate, sodium adsorption ratio, and high EC are also a reason for increased salinity stress and affect plant growth [109]. A better management approach is important to deal with salinity stress for better production. Irrigation water can be applied to maintain the salinity of soil at concentrations where higher yields can be obtained by applying excessive water to drain through the root zone and leach salts. We can apply canal water instead of underground water, if underground water is brackish. The availability of canal water is the better option to leach down salt from the root zone of crop plants. The application of soil ameliorants, such as CaSO_4_·2H_2_O, as a supplier of Ca^2+^ is the strategy for usage of unfit underground water because it improves soybean yield under saline conditions [110]. Gypsum amendment with unfit water is the best option if the availability of fit water is less than 25% [111]. It is reported that the application of CaSO_4_·2H_2_O in composition with organic manure in salt-affected soils decreases the pH, electrostatic precipitator (ESP) and EC, heavy metals, and soluble ions [112]. Liming of drainage water is also an effective strategy to reduce acidity-related constraints to increase productivity under SS [113].

### 4.2. Plant Growth-Promoting Rhizobacteria

The use of plant growth-promoting rhizobacteria (PGPR) is a useful strategy for avoiding the harmful effects of SS in legumes. In saline environments, improvement and development of salt-tolerant symbioses is important. Han and Lee [114] observed that PGPR improve photosynthesis, nutrient uptake, and growth in soybean grown under salinity. In common bean, the significant increase in shoot length was observed at salt concentrations 5.0, 7.5, and 10.0 dS m^−1^, when inoculated with *Pseudomonas chlororaphis* TSAU13 and *P. extremorientalis* TSAU20 [115]. It has been noted an increase in root and shoots growth in soybean and pea when inoculated with *P. extremorientalis* TSAU20 and *P. trivialis* 3Re27 under SS [116]. Soybean inoculation with *Pseudomonas* strains (*P. putida* TSAU1 and *Bradyrhizobium japonicum* USDA110) enhance root and shoot growth under saline environments [116]. The inoculation of soybean with arbuscular mycorrhizal fungi improves growth under SS [117]. Faba bean inoculation with *Pseudomonas fluorescens*, *P. putida,* and *Bacillus subtilis* improve growth under saline conditions [118]. Chaudhary and Sindhu [119] observed improvement in nodulation and plant biomass under SS, when inoculated with rhizobacterial and ACC^+^
*Mesorhizobium* in chickpea.

Under saline conditions, PGPR can improve legumes symbiosis with *Rhizobium spp*. [120,121]. The *P. extremorientalis* TSAU20 and *P. trivialis* 3Re27 strains improve salt tolerance in *Galaga officinalis* L. The combined application of *Pseudomonas* and *Rhizobium* strains can enhance the yield of mungbean [122]. Similarly, the combined application of PGPR and *Rhizobium* spp. improve seedling growth, nodulation, and improve osmotic stress in mungbean under salinity [123,124]. PGPR can use a number of mechanisms to improve plant growth such as producing antifungal metabolites, phytohormones, lytic enzymes, reducing ethylene production, increasing nutrient uptake, and inducing systemic resistance in plants [125,126]. In conclusion, PGPR can improve plant growth by maintaining osmoregulation, improving photosynthesis, improving nodulation, and root growth for better water and nutrient uptake, and producing phytohormones under saline conditions.

### 4.3. Seed Priming

In grain legumes, two stages, seed germination and seedling development, are critical for the establishment and most sensitive to salinity stress. Previously, a growing number of reports studied the effects of seed priming under salt stress conditions. Seed priming is a controlled hydration process that is followed by redrying and activates many of the physiological processes associated with the early phase of germination and prepares the seed for radicle protrusion [127]. Furthermore, it can decline the physical resistance of the endosperm during imbibition and repair membranes but also lead to the development of immature embryos and leach emergence inhibitors [128]. Generally, several seed priming techniques, which include osmopriming, hydropriming, chemical priming, hormonal-priming, nutrient priming, and redox priming, are adapted to induce pre-germination changes [127,129]. In a most recent report, Dai et al. [130] observed that priming with CaCl_2_, ZnSO_4_, GA_3_, and betaine hydrochloride to improve salt tolerance in soybeans seedlings. Harris et al. [131] reported that seed priming with ZnSO_4_ enhance grain yield in chickpea. It has been observed that priming with Gibberellins can enhance seed germination and improve salt tolerance of alfalfa by improving antioxidant activities and reducing membrane damage [132]. Azooz [133] observed that priming with salicylic acid enhance salinity tolerance in two faba bean genotypes. Halopriming improves salt tolerance in mungbean [134] and chickpea [135]. Hydropriming improves seedling growth, and shoot and root biomasses compared with controls [130]. Ahmadvand et al. [136] observed that seed priming with potassium nitrate enhance germination, plant height, and plant dry weight in soybean. Khomari et al. [137] reported that biopriming with *Trichoderma harzianum* improve emergence rate of seedlings, and root and shoot length in soybean. Seed priming with KNO_3_ can be improved physiological characteristic and yield in chickpea [138].

### 4.4. Role of Polyamines in Salinity Tolerance

Polyamines have relatively low molecular weight, aliphatic polycations that are widely distributed in all living organism from unicellular to multicellular organisms [139]. Three main polyamines in plant species namely: putrescine, spermidine, and spermine, even though other kinds of polyamines can also be seen in plants, for instance, cadaverine. Polyamines’ biosynthetic pathways have been studied by various researchers [140,141]. They are involved deliberately in a range of plant mechanisms, linked to plant growth and development process, for example, embryonic competence [142], fruit ripening [143], programmed cell death [144], and xylem differentiation [145]. Mounting evidence regarding polyamines suggests that they also have appropriate adaptive functional response under abiotic stresses and this is acclaimed by the considerable variation behavior in PA levels during stress. This variability in polyamine had been seemed in a number of plant species that were subjected to complex abiotic condition comprised of drought, salinity, low temperature, and low nutrient deficiency. Gu-wen et al. [146] reported that putrecine plays positive role in salinity tolerance through decreasing oxidative damage in soybean (*Glycine max*. L.). Similarly, Sheokand et al. [147] observed that the application of putrecine helps in scavenging superoxide radical as it enhanced the activity of SOD under salinity stress in chickpea. In faba bean, Suleiman [148] reported that application of putrecine regulates ion distribution in shoot and root. The application of spermidine and spermine helps to alleviate the lethal effects of salt stress and improve salinity tolerance in faba bean [149]. Radhakrishnan et al. [150] reported that the addition of spermidine significantly ameliorated the devastating effects of osmotic stress by regulating the level of plant hormones and antioxidants. In a most recent study in alfalfa, the exogenous application of sperimidine positively alleviates SS-induced damage [151]. Nahar et al. [152] reported that exogenous application of polyamines maintained nutrient homeostasis and reduced cellular Na content as well as regulated endogeneous polyamine concentrations in mungbean under salinity stress. In another study, the application of spermine confers salt tolerance in mungbean [153].

### 4.5. Selection and Conventional Breeding Approaches

In the case of complex polygenic traits, like tolerance to salinity, bulk and recurrent selection is adopted in order to increase the frequency of minor genes with an additive effect. From both physiological and genetic aspects, SS tolerance is a complex trait. In legume crops, economically feasible screening approaches for SS tolerance are lacking. In order to develop legumes with SS tolerance, an integrated approach comprising using existing genetic variation, exploiting diverse and novel sources to create new variations, and using breeding strategies with several traits instead of a single trait may be effective [154,155,156]. In legumes, mass screening is frequently adopted to select salt-tolerant germplasm to develop better performing legume genotypes. Sehrawat et al. [156] reported 117 genotypes of mungbean for SS tolerance and noted decreased seed germination and seedling growth in all 117 genotypes, but all 117 genotypes performed differently under SS and were categorized as moderately tolerant, highly tolerant, tolerant, highly susceptible, moderately susceptible, and susceptible genotypes. To screen for SS tolerance various traits/characters have been used such as seedling emergence, leaf soluble proline, leaf Ca^2+^/Na^+,^ and K^+^/Na^+^ ratio, stomatal conductance, photosynthetic activity, nodulation, osmotic adjustment, pods per plant, and yield. To screen genotypes for salinity stress tolerance, ion homeostasis is another important trait being used; however, the involvement of individual Cl^−^ or Na^+^ exclusion to SS tolerance has not been confirmed in certain legumes, like chickpea [23].

The accumulation of Cl^−^ and Na^+^ to lethal concentrations in the reproduction phase in legumes (*Cicer arietinum* L.) made it a highly sensitive phase to salinity [18]. However, no correlation was observed among Na^+^ accumulation in shoots and yield due to SS in chickpea [57]. It appears that for legume crops like chickpea, a combination of mechanisms, e.g., tissue tolerance of excess ions and ion exclusion, plays an important role under saline conditions. However, in mungbean, breeding approaches need to comprise diverse background parental lines to produce genetically engineered salt-tolerant genotypes [16]. Since the ultimate aim for SS tolerance is yield under stress, the traits/characters used for evaluating SS tolerance must be correlated with yield [18] or genotypes selected for yield under saline conditions. To develop salt-tolerant genotypes of legumes, direct selection has been conducted at several locations [157]. For salinity tolerance, mass screening can be achieved on the basis of osmotic adjustment, homeostasis, plant biomass, and grain yield under saline conditions.

It has been known for decades that salinity stress appears to be a polygenic and quantitative trait in nature and regulated by several genes under diverse environments [158]. In order to improve understanding about salinity stress on a genetic basis, significant efforts have been made to identify different quantitative trait loci (QTL). The adoption of genomic and genetic analysis in several crops to identify DNA regions strongly associated with such quantitative traits, the molecular markers, to assist in breeding methods [159]. Moreover, an increasing number of previously published reports suggested the essential role of these markers for the indirect screening of improved crops speed up the screening process by alleviating laborious and time-consuming of direct selection in field and greenhouse. Previous studies also reported that DNA markers, simple sequence repeats (SSR), restriction fragment length polymorphism (RLFP), random amplification of polymorphic DNA (RAPD), and amplified fragment length polymorphism (ALFP) for abiotic stress [160]. In *C. arietinum*, various SSRs were identified to have a strong linkage with both seed yield under salt stress and control conditions [57]. An attempt to map QTL for SS tolerance, by using recombinant inbred lines (RIL) from a cross among salt-tolerant JG62 and salt-sensitive ICCV2 chickpea lines, identified grain yield varying during salinity, with a number of grains being the most closely related trait to yield [161]. A number of QTL were reported for grain yield and yield components during salt stress within each phenology group, but no major QTL was found when an analysis was made on the whole RIL population. The recent advance in genome-based approaches have endorsed the development of high-throughput approaches for genotyping, allowing the finding and accessing desirable alleles, different QTL having the potential to affect desired responses. In field pea, single nucleotide polymorphism (SNP) markers associated with expressed sequence tags (ESTs) were developed and used to generate comprehensive linkage maps for field pea [162]. They reported that out of 768 variant nucleotide positions screened for genotyping of RIL population, a total of 705 SNPs successfully detected segregating polymorphisms. More importantly, identifying new QTL plays a key role in crop improvement through marker-assisted selection (MAS). Furthermore, many breeding methodologies have been applying in the improvement of salt resistance in grain legumes based on MAS. MAS are often used to separate desirable QTL by mapping, to identify linkage drag correlated with undesirable alleles linked to target genes. A growing number of reports showed that salt tolerance is controlled by a minor QTL in chickpea [163], two minor QTL in field pea [162], single major QTL in soybean [160] and several minor QTL in *Medicago truncatula* [162]. In a most recent study, Shi et al. [164] identified a major salt tolerant QTL, which was flanked by SSRs Barcsoyssr_03_1421 and GMABAB on chromosome 3 in soybean, single trait composite interval mapping, based on single-marker regression, and multiple interval mapping analysis. A recent report has showed that QTL mapping for SS resistance in a *Vigna* species, cowpea known to be halophytic in nature [165]. Chankaew and co-workers built a genetic map, consisting of 150 SSRs, from F2 population derived from a cross among *V. marina subsp* and *V. luteola*. Evaluation of F_2:3_ populations for salinity tolerance in hydroponic conditions at seedling and developmental stages, and segregation analysis showed that salinity tolerance in *V. marina* is regulated by a less number of genes. Approximately 50% phenotypic variation could be explained by multiple interval mapping which has regularly identified one major QTL. The flanking markers may assist to transfer allele that has tolerance against salt from *V. marina* subsp. *oblonga* into related *Vigna* crops. However, by implementing the MAS breeding method, which has certain limitations in terms of unreliability, marker validation, hindrance in association, inaccessibility of marker variation among the population, linkage drag, and breeding incompatibility problems among species [166], there is an urgent need for alternative molecular breeding to minimize the drawbacks linked to MAS approach to pave the way to comprehensively understand the integration of functional, structural, and comparative genomics.

### 4.6. Bioengineering and Functional Genomics

In legume crops, SS-induced genes are well distributed all over the genomes. QTL identification of related traits/characters in combination with marker tagging has turned out to be an important medium for the pointed insertion of the required trait to the unadapted trait. In recent times, crop improvement through molecular marker has gained momentum, where whole-genome sequencing (WGS) has built a strong foundation of newly developed single-nucleotide polymorphism (SNP) markers, simple sequence repeats (SSRs), and next-generation sequencing (NGS) technology, enhancing its cost-effectiveness and efficiency. The capability of molecular markers to screen QTL has been explored to develop crop varieties with improve SS tolerance under saline conditions. Guan et al. [167] screened 58 soybean accessions with three DNA markers, Barcsoyssr-3-1306, Barcsoyssr-3-1310 and QS080465 (InDel marker), which co-segregated with the salt-tolerance locus, when crossing salt-tolerant (Tiefeng 8) and salt-sensitive (85–140) soybeans. Additionally, Lee and coworkers studied the association among the linked SSR markers (Sat-091 and Satt-237) and salt-tolerance QTL by tracing the pedigrees of FT-Abyara and S-100, the parents used in QTL mapping [160]. The DNA markers associated with QTL might be useful for marker-assisted selection to pyramid tolerance genes for both saline and alkaline stresses [168]. However, QTL mapping would be highly efficient and effective if linked genes along with the computational improvements were added to the analysis. Sehrawat and coworkers developed 38 SSRs (novel microsatellite markers) for the detection of genetic variations in 12 mungbean genotypes (nine cultivated and three wild) under salt stress [156]. The developed SSRs help to explore the genetic variations in mungbean cultivars having a narrow genetic base, as well as within related legume crops [169]. Marker index (MI) and high-resolution power (RP) further confirmed that these SSR markers are highly helpful. Therefore, the wild mungbean can be selected for the improvement of the genotypes for SS tolerance by broadening their genetic base. Assessment of the diversity among the examined genotypes would be of significance for designing breeding strategies for the improving quantitative trait alleviating SS tolerance [170]. The SSRs are very helpful in the identification of candidate genes or QTL capable of improving salinity tolerance. The SSRs associated to the genes/trait may assist for quick selection of genotypes rather than phenotypic screening in breeding programs [171]. Under drought stress, the addition of ABA or mi-RNAs accumulation has been studied in beans, but their function under saline conditions has not been investigated [172]. However, up-regulation of mi-RNAs play an important role in managing salinity stress in soybean [173].

Transferring genes from one crop to other crops to achieve required traits (qualitative or quantitative) is termed as the transgenic approach and this approach is very effective and efficient compared to conventional breeding. Several key SS-induced genes, including *MiR172C*, *GmbZIP132*, *GmMYB177, GmMYB92*, *GmMYB76*, and *GmSALT3*, have been reported, and their related mechanisms were studied [174,175,176]. Guan et al. [176] reported an SS-induced gene (*GmSALT3*) located on the chromosome 3 and has a dominant role in limiting Na^+^ accumulation in leaves and shoots of soybean. Certain candidate genes for SS tolerance have been identified in legumes (Table 2). Neng et al. [39] reported that *GmNHX1* and *GmNcl1* might play important role in alleviating SS tolerance in soybean. Xue et al. [177] reported that overexpression of *GmGMP1* conferring tolerance to high salinity stress during seed germination in soybean and *Arabidopsis*. Chen et al. [178] suggested that *GmSK1* might play an important role to improve SS tolerance in plants. Li et al. [179] observed that overexpression of *GmNAC15* in soybean hairy roots improve SS tolerance. An et al. [180] suggested that *PgTIP1* is an SS tolerance associated gene involved in improving SS tolerance in transgenic soybean lines. Overexpression of *GmBIN2* enhances SS tolerance in soybean and *Arabidopsis* [181]. The expression of *AtB7*, *AtBF3*, and *AtDREB2A* genes improves salinity tolerance through the accumulation of GB and proline in peanut [182]. Lopez et al. [183] observed that GB improves relative leaf water content and stomatal conductance in beans under SS. A number of genes have been successfully transferred into several species of legume to enhance SS tolerance by accumulating and synthesizing osmolytes, the uplifting activity of antioxidants, and improving sodium vacuolar sequestration. *MtZpt2-1* is an SS tolerance associated gene involved in improving SS tolerance recovery of root growth and the development of root nodules after the salinity stress [184]. Additionally, the *PR10a* gene played a vital role in alleviating salinity tolerance in faba bean [185]. Similarly, transgenic lines of chickpea with the AP2-type TFs, CAP2, improve SS tolerance [184]. In alfalfa, Bai et al. [186] observed that *GsCBRLK* enhanced SS tolerance by improving chlorophyll content and by lowering malondialdehyde (MDA) content and membrane leakage. Likewise, overexpression of *GsCBRLK* in soybean improves salinity tolerance by enhancing photosynthesis and scavenging ROS [187]. In another study, overexpression of *TaNHX2* improved salinity resistance in transgenic soybean plants [188]. Bao et al. [189] also reported enhanced tolerance to salt in transgenic alfalfa due to the overexpression of *Arabidopsis* H^+^Pase. The expression of genes induced by SS and its regulation comprise a range of molecular mechanisms that are controlled via several TFs in various plant species. These TFs regulate and initiate genes transcription or gene products specified for salinity tolerance. Therefore, the role of transcriptomics approaches in gene regulation, its expression, and identification of genes involved in salinity tolerance has been useful in recent times. In legumes, various TFs have been identified to impart salinity tolerance, such as GmNACs, GmAP2/ERFs, and GmTDF-5 in soybean, MtNAC969 in barrel clover, GsZFP1 and AtAvp1 (H^+^-PPases) in alfalfa [190,191]. Tripathi et al. [191] reported a number of families of TFs, e.g., AP2/ERF, WRKY, bHLH, and NAC, which have been linked with SS tolerance in groundnut. Similarly, transgenic lines of chickpea with the putative NAC-type TF, CarNAC4, improve salinity tolerance by decreasing MDA concentration [192]. Sarkar et al. [193] reported TF (*DREB1A*) in groundnut plants confer SS tolerance at the seedling stage. Similarly, the *DREB1A* gene played a similar function in alleviating salinity stress tolerance in lentil [194].

Mutation breeding has been extensively used to develop salt-tolerant legumes but it is a laborious approach as several plant generations are required to identify the mutation of the targeted genome [195]. The WGS approach has made it a faster and easy method as mutations can be identified and mapped simultaneously [196]. The identification of genes induced by SS using WGS and SNP technologies may be useful to develop stress-tolerant genotypes and understanding of molecular mechanisms of salinity tolerance [197]. The NGS can also be used to study differential gene expression in saline conditions enabling the identification of target-induced mutations in fewer generations [198,199,200].

### 4.7. CRISPR-Cas9: Master Player for Genome Editing (GE)

Genome editing (GE), as the name implies, is the targeted mutagenesis of the genome. Current tools allow us to introduce specific modifications at target sites in the genome. Due to comprehensive genomic research, plenty of information is available about salinity tolerance in plants. The role of small RNAs, proteins, and different genes under stress environments is well-defined. The genetic transformation of plants for enhancing salinity tolerance and increasing yield is successful up to some extent [217]. This has been done by adopting biotechnological approaches, such as screening, cloning, overexpression, and crossing, which are time consuming and laborious. To tackle these issues, novel GE tools were introduced. Targeted mutation, precise sequence modification, deletion, or insertion can be achieved by using these tools. At present, CRISPR-Cas9, Transcriptional activator-like effector nuclease (TALENs), and zinc finger nuclease (ZFNs) are available for GE [218]. Out of these three tools, CRISPR-Cas9 can manipulate any genome sequence to study its function [219] (Figure 4). These tools’ applications will lead to the development of non-genetically modified crop plants with the wanted trait that can contribute to improved crop productivity under stress environments [220]. In recent years, GE by CRISPR/Cas9 has been observed in many crops, including rice [221], wheat [222,223,224], maize [225,226], barley [227,228], rapeseed [226], potato [228], sweet orange [229], soybean [230,231,232], poplar [232], and petunia [233]. Abdelrahman and co-workers reviewed the targeted mutagenesis in crop plants by CRISPR/Cas9. They report CRISPR/Cas9-mediated GE in many crops for yield improvements of crop plants grown under unfavorable conditions [234]. Mushtaq et al. [235] reviewed recent applications of the CRISPR/Cas9-mediated GE as a means to develop crop plants with increased tolerance to the abiotic stresses they encounter when grown under unfavorable conditions. CRISPR- Cas9 is a powerful tool in engineering for salt tolerance in legumes but fewer investigations have been done previously in GE for SS tolerance in legumes. The use of targeted GE tools, especially CRISPR/Cas9, has great potential to develop high-yielding legumes under saline conditions.

## 5. Conclusions and Future Research Perspectives

Crop production is adversely affected by SS. It is quite evident that SS significantly affects legume yield and currently has become a prime concern for crop production. Legumes undergo various adaptations, for instance, osmotic adjustment and osmoregulation, activation of antioxidant defense systems, hormonal regulation, and mechanisms, such as lethal ion elimination, to survive under SS. There is an immense need to develop new legume varieties with stable and higher yield across unfavorable environments. Availability of legume genomes will aid in the identification of genes that could be deployed for use in transgenic or breeding approaches to develop salt-tolerant legumes. Recent investigations identified molecular markers, QTL, and genes associated with stress tolerance which can help to improve growth and yield under saline conditions. This review not only provides knowledge and understanding about the effect of salinity stress on legumes but also gives us knowledge about the mechanisms and management strategies to improve legume yield. Although the investigation of SS responses has progressed well in cereals and other plants, it still has a long way to go in legume crops. Identification of SS-induced genes in several legume crops, in-depth study of downstream and upstream elements, and comprehensive study of gene expression at different developmental stages with advanced tools will help in elucidating and understanding this trait. Bioinformatics has emerged as a useful tool to make the omics data available to researchers through the establishment of public databases. Hence, there is a need to create databases comprising information on metabolomics, ionomics, and phenomics, particularly in grain legumes. Recently, speed breeding (SB) technique gained attention of researchers because it enables growing up to six plant generations in a single year. The integration of SB with other advanced crop breeding approaches, including GE, high-throughput genotyping, and genomic selection, accelerating the rate of crop improvement under unfavorable conditions [237]. In the future, integration of several approaches, such as genomic, transcriptomic, proteomic, metabolomics, and agronomic strategists, as well as advanced GE tools (CRISPR- Cas9) is warranted to develop legume varieties with high salt tolerance under saline conditions (Figure 5).

## Figures and Tables

**Figure 1 ijms-20-00799-f001:**
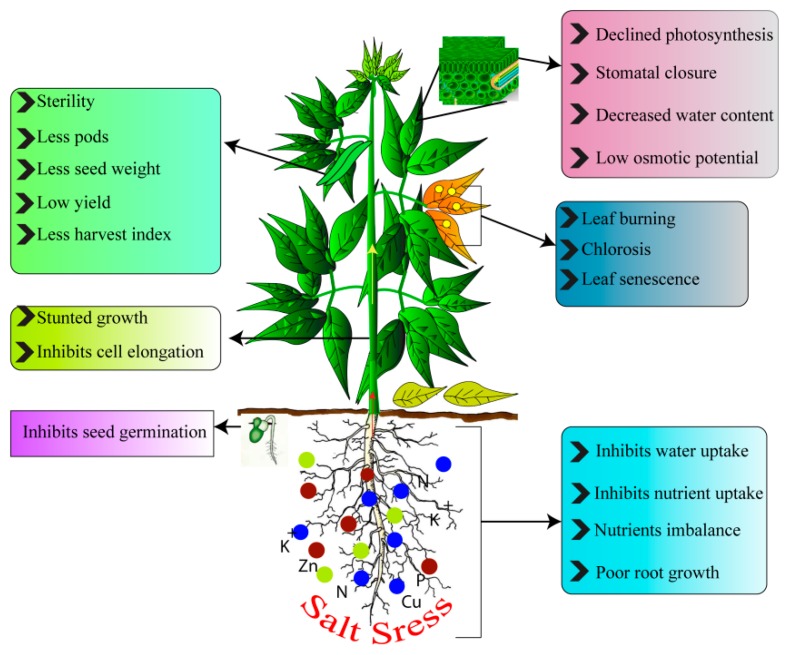
Schematic representation of plant response to salt stress (SS).

**Figure 2 ijms-20-00799-f002:**
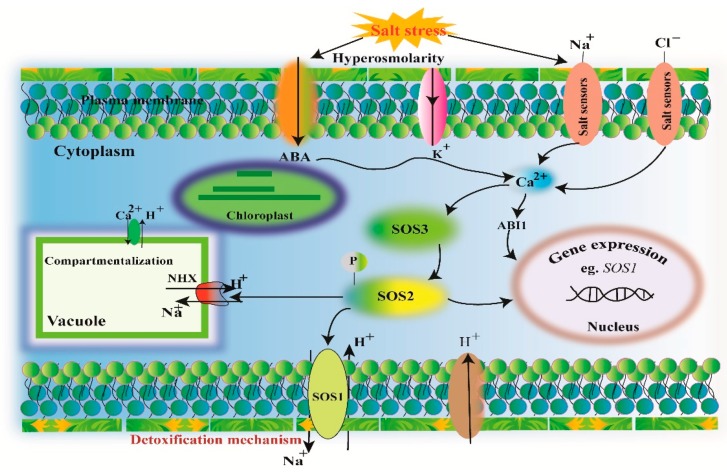
Schematic representation of the salt overly sensitive (SOS) pathway for salinity stress response. The SOS pathway regulates the Na^+^/H^+^ antiporters. The SS-induced increase in Ca^2+^ concentration in cytosol and is sensed by SOS3. SOS3 work together with SOS2 and triggers its kinase activity. The SOS2–SOS3 localized in plasma membrane. Then SOS2 phosphorylates SOS1 and triggers its antiporter (Na^+^/H^+^) activity assisting Na^+^ efflux from plant cell.

**Figure 3 ijms-20-00799-f003:**
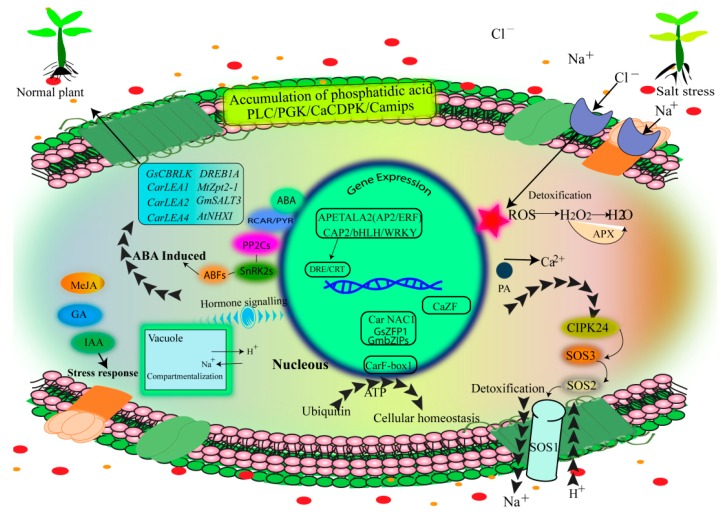
Schematic representation of salinity tolerance mechanism in Legumes. Reactive oxygen species (ROS), Ca^2+^ and ABA are activated under SS. SS induces synthesis of ABA, which, in turn, upregulates the transcription of ion transporter genes. Overexpression of transcription factors (*GmbZIPs*, *GmNACs*, *GsZFP1*, *AP2*/*ERF*, *CarF* box-1, and *CarNAC*, *CAP2*) have been reported under salinity.

**Figure 4 ijms-20-00799-f004:**
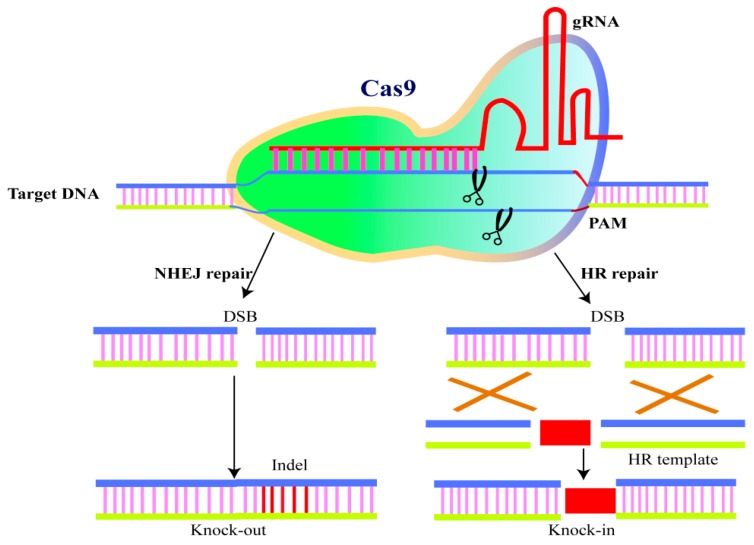
Schematic representation of Cas9/gRNA genome editing. CRISPR/Cas9 system consists of sgRNA and Cas9 nuclease. The sgRNA directs the sequence-specific Cas9 nuclease to start double-strand DNA breaks in the target DNA. The cell’s DNA repair machinery, comprising HR and NHEJ pathways, repairs the strand breaks, creating a short deletion/insertion (gene knockout), new sequence insertion and/or sequence modification. sgRNA, single guide RNA; HR, homologous recombination; NHEJ, non-homologous end joining (adopted and modified from [236]).

**Figure 5 ijms-20-00799-f005:**
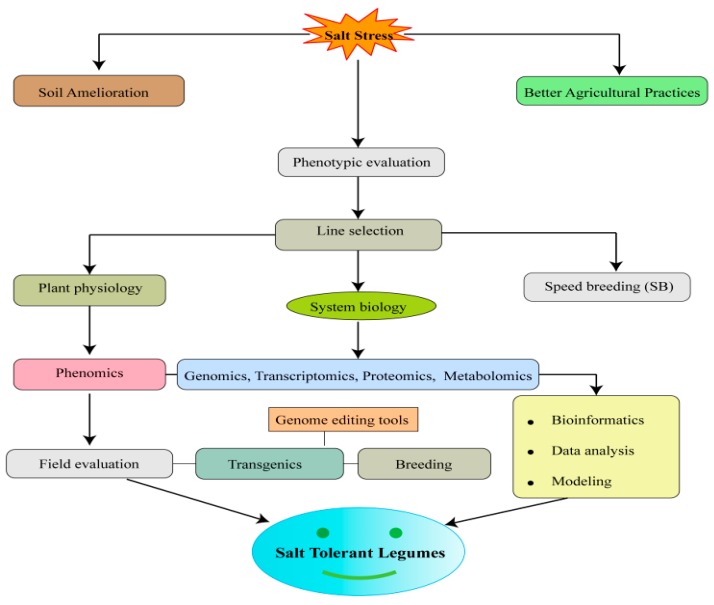
Schematic representation of approaches for developing of salt tolerant legumes. The integration of “omics” approaches along with advance genome editing (GE) tools and agronomic approaches can improve salt tolerance in legumes.

**Table 1 ijms-20-00799-t001:** Yield reduction under different salinity levels.

Legume Crops	Salt Concentration	Yield Loss (%)	Reference
Soybean (Galarsum)	14.4 dSm^−1^	50%	[54]
Soybean (Lee)	8.5 dSm^−1^	53%	[50]
Soybean (loam soil)	7 dSm^−1^	46%	[20]
Soybean (caly soil)	6.3 dSm^−1^	46%	[62]
Mungbean (cv. Pusavishal)	50 mM NaCl	41%	[32]
Mungbean (var. 245/7)	8 dSm^−1^	60%	[58]
Mungbean (var. NM-51)	12 dSm^−1^	77%	[58]
Mungbean (var. NM-92)	8 dSm^−1^	61%	[58]
Mungbean (var. 6601)	12 dSm^−1^	72%	[58]
Chickpea (var. FLIP 87-59)	3.8 dSm^−1^	69%	[62]
Chickpea (var. FLIP 87-59)	2.5 dSm^−1^	43%	[20]
Chickpea (var. ILC 3279)	3.8 dSm^−1^	72%	[20]
Fababean (loam soil)	6.6 dSm^−1^	50%	[62]
Fababean (clay soil)	5.6 dSm^−1^	52%	[62]
Fababean (loam soil)	4.9 dSm^−1^	28%	[20]
Fababean (clay soil)	4.3 dSm^−1^	19%	[20]
Lentil (cv. 6796)	3.1 dSm^−1^	100%	[20]
Lentil (cv. 6796)	2 dSm^−1^	14%	[20]
Lentil (cv. 5582)	2 dSm^−1^	24%	[20]
Lentil (cv. 5582)	3.1 dSm^−1^	88%	[20]

**Table 2 ijms-20-00799-t002:** Transgenic legumes for better salt tolerance.

Transgenic Crop	Gene Transferred	Source	Function	Reference
Soybean	*P5CS*	*Solanum torvum* Sw.	Synthesis and accumulation of proline	[200]
	*TaNHX2*	*T. aestivum* L.	Sodium vacuolar sequestration	[188]
	*WRKY11*	*Medicago sativa* L.	Improves salt tolerance	[201]
Pea	*Na^+^/H^+^*	*Arabidopsis thaliana* L.	Sodium vacuolar sequestration	[202]
	*P5CS*	*Arabidopsis thaliana* L.	Synthesis and accumulation of proline	[203]
Chickpea	*P5CS*	*Vigna aconitifolia* L.	Synthesis and accumulation of proline	[204]
Faba bean	*PR10a*	*Solanum tuberosum* L.	Synthesis and accumulation of osmolytes	[185]
Mashbean	*gly I*	*Brassica juncea* L.	Increase in antioxidant ability	[205]
Pigeon pea	*VaP5CSF129A*	*Vigna aconitifolia* L.	Synthesis and accumulation of proline	[206]
	*VaP5CSF129A*	*Vigna aconitifolia* L.	Synthesis and accumulation of osmolytes	[207]
Lentil	*DREB1A*	*Arabidopsis thaliana* L.	Synthesis and accumulation of osmolytes	[194]
Peanut	*AtNHXI*	*Arabidopsis thaliana* L.	Sodium vacuolar sequestration	[208]
	*AtDREB1A*	*Arabidopsis thaliana* L.	Improves salt tolerance	[193]
	*AtHDG11*	*Arabidopsis thaliana* L.	Improves salt tolerance	[209]
Alfalfa	*SsNHX1*	*Suaeda salsa*	Regulate plant Na^+^/H^+^ antiporters	[210]
	*CsALDH12A1*	*Cleistogenes songorica* L.	Improves salt tolerance	[211]
	*GmDREB1*	*Glycine max* L.	Conferred salt tolerance	[212]
	*IbOr*	*Ipomoea batatas* L.	Increased tolerance to multiple abiotic stresses	[213]
	*GsCBRLK*	*Glycine soja* L.	Improves salt tolerance	[186]
	*TaNHX2*	*Triticum aestivum* L.	Regulate plant Na^+^/H^+^ antiporters	[214]
	*ScNHX1* and *ScVP*	*Suaeda corniculata*	Vacuolar membrane H^+^-pyrophosphatases and H^+^-ATPases	[215]
	*GsZFP1*	*Glycine soja* L.	Improves salinity tolerance	[216]

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
