# Peer review of "Grain Legumes and Fear of Salt Stress: Focus on Mechanisms and Management Strategies"

_ijms, 2019, doi:10.3390/ijms20040799_

Round 1
Reviewer 1 Report
Addition of table.
Please add a table with these parameters.
1. Legume name and cultivar (Include one or two most salt tolerant cultivars /variety under each legume)
2. Salt Sensitivity levels in ds/m
3. % yield loss
4. Reference
Please add a separate diagrammatic figure of SOS pathway. Discussion is sufficient.
Please discuss details of grain yield loss and reduction in quality of major legumes under salt stress.
Please discuss how nitrogen fixation is affected under Salt stress. Discuss salt levels at which nitrogen fixation is affected.
Please add and discuss the role of polyamines in salt stress tolerance.
Include details on Molecular markers and QTLs in conventional breeding
Include benefits of seed priming and compounds used for priming for improving salt stress tolerance in legumes.
Author Response
Dear Reviewer,
We thank you for your thoughtful and supportive comments. We have carefully considered each comment and extensively revised the manuscript accordingly. The following are our answers and explanation to each comment.
Reply to Reviewer # 1:
Comments and Suggestions for Authors
Point 1: Addition of table.
Please add a table with these parameters.
1. Legume name and cultivar (Include one or two most salt tolerant cultivars /variety under each legume)
2. Salt Sensitivity levels in ds/m
3. % yield loss
4. Reference
Response 1: Thank you for your comment. We have added a table in the revised manuscript according to your comment.
Point 2: Please add a separate diagrammatic figure of SOS pathway. Discussion is sufficient.
Response 2: Thank you for your comment. We have added a separate diagrammatic figure of SOS pathway in the revised manuscript according to your comment.
Point 3: Please discuss details of grain yield loss and reduction in quality of major legumes under salt stress.
Response 3: Thank you for your comment. We have revised it in the revised manuscript according to your comment.
Point 4: Please discuss how nitrogen fixation is affected under Salt stress. Discuss salt levels at which nitrogen fixation is affected.
Response 4: Thank you for your comment. We have revised it in the revised manuscript according to your comment.
Point 5: Please add and discuss the role of polyamines in salt stress tolerance.
Response 5: Thank you for your comment. We have added and discuss the role of polyamines in salt stress tolerance in the revised manuscript according to your comment.
Point 6: Include details on Molecular markers and QTLs in conventional breeding
Response 5: Thank you for your comment. We have revised it in the revised manuscript according to your comment.
Point 7: Include benefits of seed priming and compounds used for priming for improving salt stress tolerance in legumes.
Response 5: Thank you for your comment. We have included a section about seed priming in the revised manuscript according to your comment.
Thank you for your consideration again.
Regards,
Prof. Xiaobo Wang, corresponding author
Reviewer 2 Report
The review titled: “Grain Legumes and Fear of Salt Stress: Focus on Mechanisms and Management Strategies” described the impact of salt stress (SS) on legumes. The authors described the recent findings in crops tolerance, adaptation and the several approaches used to improve SS tolerance in legumes.
In my opinion, the review is fairly well written and the topics are well structured. However, there are a few minor changes that need to be done, in order to improve the quality of the MS.
See below my comments:
Point 1: Page 1 line 36: The sentence “Legumes belong to family Fabaceae and are a nourishing and low-cost food belongs to family Fabaceae.” There is repetition (belong to family Fabaceae) in this sentence.
Point 2: Page 2 line 69: The scientific name (Phaseolus) is not italic.
Point 3: Check better the word typing in figure 1 and improve figure 1.
Point 4: Page 3 line 117: Did you mean avoiding?
Point 5: Page 3 line 148: Phase I, not PhaseI.
Point 6: Page 5 line 187: Abbreviation of ABA (Abscisic acid) is not used at first occurrence.
Point 7: Page 6 line 239: Correct spellings of additionally.
Point 8: Page 7 line 272: The Abbreviation ESP occurs. What is it and what does it stand for?
Point 9: Page 7 line 277: I didn’t find Han and Lee, (2005) in the reference section.
Point 10: Page 7 line 285: The name is Faba bean, not feba bean.
Point 11: Page 7 line 287: The Abbreviation ACC occurs. What is it and what does it stand for?
Point 12: Page 7 line 289: Rhizobium spp. needs to be italic.
Point 13: Page 8 line 349: The name is Mungbean not mung beam.
Point 14: Page 11. The Abbreviations (HR and NHEJ) used in Figure 3. What is it and what does it stand for?
Point 15: Page 12. Check better the word typing in Figure 4.
Author Response
Dear Reviewer,
We thank you for your thoughtful and supportive comments. We have carefully considered each comment and revised the manuscript accordingly. The following are our answers and explanation to each comment.
Response: Thank you for your comment. We have carefully revised all minor mistakes it in the revised manuscript according to your comment and improve the manuscript.
Regards,
Prof. Xiaobo Wang, corresponding author